# Finding the 'sweet spot' between customisation and workflows when optimising ePrescribing systems: a multisite qualitative study

Catherine Heeney [1], Stephen Malden [2], Aziz Sheikh[3]

¹Centre for Medical Informatics, Usher Institute, The University of Edinburgh, Edinburgh, UK
²Advanced Care Research Centre, Usher Institute, The University of Edinburgh, Edinburgh, UK
³Division of Community Health Sciences, Usher Institute, The University of Edinburgh, Edinburgh, UK

**Correspondence to**
Dr Stephen Malden;
stephen.malden@ed.ac.uk

## ABSTRACT

**Objectives** The introduction of ePrescribing systems offers the potential to improve the safety, quality and efficiency of prescribing, medication management decisions and patient care. However, an ePrescribing system will require some customisation and configuration to capture a range of workflows in particular hospital settings. This can be part of an optimisation strategy, which aims at avoiding workarounds that lessen anticipated safety and efficiency benefits. This paper aims to identify ePrescribing optimisation strategies that can be translated into hospitals in different national settings. We will explore the views of professionals on the impact of configuration and customisation on workflow.

**Design** This paper draws on 54 qualitative interviews with clinicians, pharmacists and informatics professionals with experience of optimising ePrescribing systems in eight hospital sites and one health system, in four different countries. Interview transcripts were analysed using an inductive thematic analysis.

**Setting** Secondary and tertiary care hospitals in the UK, USA and mainland Europe.

**Participants** Fifty-four healthcare workers with expertise in clinical informatics.

**Results** Five identified themes following thematic analysis showed that optimisation of ePrescribing systems can involve configuration and/or customisation. This can be a strategy to combat workarounds and to respond to local policy, safety protocols and workflows for particular patient populations. However, it can result in sites taking on responsibility for training and missing out on vendor updates. Working closely with vendors and other users can mitigate the need for extensive system modification and produce better outcomes.

**Conclusions** Modifying an ePrescribing system remains key to enhancing patient safety, and better captured workflow remains key to optimisation. However, we found evidence of an increasingly cautious approach to both customisation and configuration among system users. This has led to users seeking to make less changes to the system.

## STRENGTHS AND LIMITATIONS OF THIS STUDY

⇒ The sample included in this study is comprised of healthcare workers from institutions with extensive experience in the customisation of ePrescribing systems.
⇒ The study sampled from hospitals across different international locations, accounting for differences in policy contexts.
⇒ We focus only on Organisation for Economic Cooperation and Development countries, this means that we will have failed to capture interesting examples of ePrescribing beyond that.
⇒ Some types of system are over represented, in particular integrated systems provided by Epic and Cerner were present in the majority of our sites.

## INTRODUCTION

Around 10% of preventable harms to inpatients is attributable to errors in the prescribing process.[1] Electronic health record (EHR) systems offer the potential to improve the safety of prescribing and medication management decisions and patient care.[2] The introduction of such systems has the potential to minimise medication error.[3] However, it is also acknowledged that an EHR will not be perfectly adapted to capture existing workflows in a given hospital and will require fine tuning to local safety protocols as well as professional and specialisation needs.[4 5] Developing the ePrescribing system in situ is key to making it safe to use, in a 'high risk' setting such as healthcare.[6] In the complex and evolving secondary care context, it is unrealistic to imagine that commercial off-the-shelf (COTS) systems can offer optimal performance with local or indeed wider organisational requirements immediately post implementation.[7 8]

Changes to an ePrescribing system can take the form of configuration and customisation. Configuration works with existing options available in the system or from the vendor by changing and refining rules to reflect processes and practices in local and national settings.[7] Customisation involves more fundamental coding changes or 'modifying the

underlying software to improve functionality'.[9] Hospitals resort to this when the vendor does not provide sufficient configurability to capture necessary workflows, or where it is more financially viable to configure/customise existing systems than it is to purchase additional systems.[10]

Workarounds arise where there is a mismatch between practices within particular hospitals and the functionalities and capacities of the ePrescribing system.[4 5 11–14] They remain a problem especially when they by pass in built safety features.[1 6 7] Configuration and customisation may look like an obvious win in terms of better capturing workflow. Moreover, this route can be necessary where, for example, there are particular patient populations or national policy requirements.[10] However, drawbacks of extensive configuration and customisation include being left behind for standard updates from the vendor and uncertainty as to where responsibility for making improvements and routine maintenance resides after customisation of a system has taken place.[9 10]

Modifications to reflect local requirements and system capabilities are key to system optimisation,[4 7 15] which has been defined as, 'the activity of enhancing system capabilities and integration of subsystem elements to the extent that all components operate at or above user expectations'.[16] Hospitals face an inevitable tension between an objective to offer a universal, coordinated and standardised approach to system functionality on the part of the vendor and national health systems, and a requirement to accommodate specific workflows at local level [3, 5 (10)]. The impact and learning arising from optimisation strategies, reliant on configuration and customisation, over time is still relatively unexplored in existing literature on ePrescribing in comparison to the equivalent literature on the implementation of new ePrescribing systems to replace paper-based prescribing. Below, we will discuss strategies used to incorporate and manage ePrescribing systems and workflow and the mutual process of change that this interaction produces as well as shifts to in attitudes to modifying system functionality.

## METHODOLOGY

The qualitative fieldwork described in this paper is part of a wider study on Optimisation of ePrescribing in Hospitals (eP Opt). The methods employed for this present study are outlined below; for a more detailed description, see Ref. [17]. We employed a qualitative multisite study design with semistructured interviews carried out in each of the study sites. The aim was to capture strategies and practices for optimisation of ePrescribing in hospitals, which are applicable both within the UK and internationally.

### Selection of study sites

Eight hospitals and one healthcare provider participated in this study across four countries (US, UK, Netherlands and Norway). All sites had significant experience of implementation and optimisation of EHRs and ePrescribing, Computerised Physician Order Entry and

Clinical Decision Support. All hospitals had begun digitisation at some level at least a decade before fieldwork began. We selected sites awarded Healthcare Information and Management Systems Society level 6 or 7; HIMMS is a widely used measure of digital excellence. The eight hospitals were high profile teaching hospitals. A purposive sampling strategy was used to select cases[18] (see (online supplemental table 1). We identified sites through a scoping review of optimisation strategies in ePrescribing[19] and two expert roundtable events.[17] Only Organisation for Economic Cooperation and Development (OECD) were included to increase the internal comparability of the sample (online supplemental table 1).

### Data collection

We contacted 10 hospitals in total. In all cases, we had an initial meeting with a gatekeeper(s) who helped identify relevant professionals who had been expensively involved in their respective ePrescribing system optimisations. Due to COVID-19 restrictions and lack of staff availability, one site in Spain did not take part. In the remaining sites (see online supplemental table 1), we contacted relevant professionals by email, with a consent form and information sheet. If participants were willing to participate, they retuned a signed and dated consent form via email to the corresponding researcher(s), who then arranged with them an appropriate time to conduct the interview. We interviewed 54 professionals including clinicians, Chief Information Officers (CIOs), Chief Medical Information Officers (CMIOs), pharmacists and IT and data specialists [7].

Planned site visits were replaced by remote interviews using approved online platforms, including Teams, Zoom, nhn.no and Skype. Initial contact with the sites was in early 2020, with interviews beginning in the first site in May 2020 and the final interview was conducted in May 2021. Two experienced qualitative researchers (CH and SM) conducted semistructured interviews following an interview topic guide. The topic guide was designed to investigate the wider ePrescribing optimisations undertaken at the study sites, but included specific questions relating to customisation and configuration. Specifically, participants were asked to detail any customisations/configuration that had taken place at their sites, and to summarise the reasons for this, and the perceived benefits/repercussions of such optimisations on the overall functionality of the ePrescribing systems. Interview questions were developed prior to data collection during 'expert roundtable' consultations which involved a structured workshop event. Relevant researchers, policy-makers and practitioners in the field of clinical informatics were invited to this event, to help guide the direction of the wider eP Opt project.[17] Following the event, the research team developed interview topic guides based on the direction of attending experts, and piloted the questions internally. The researchers had no prior relationship with participants. Interviews lasted between

30 and 90 min depending on the interviewees' time and availability and were recorded and transcribed verbatim.

## Data analysis

The research team (CH and SM) first independently coded two transcripts and discussed any discrepancies before finalising the coding framework (online supplemental material). The researchers then coded all 54 transcripts. Transcripts were analysed using an inductive thematic analysis and the data grouped into themes and subthemes.[20] We employed NVivo V.12 pro qualitative data analysis software. For this paper, we extracted data coded to relevant codes including configuration and workflow, which were then further categorised into five cross cutting themes.

## Patient and public involvement

A major component of the eP Opt project is the involvement of patient and public representatives across the four project phases. Specifically, two patient and public (PPI) representatives are involved as team members, who attend research meetings and public events to provide feedback and suggestions on the work within each phase from a patient's perspective. They have also been extensively involved with assisting with the design of an upcoming PPI roundtable event for the project, progress of the study has been shared with a group of invited patients, and their feedback brought to bear on research decisions and directions. These PPI consultations have helped in the formulation of research questions and fed into analysis of the data by highlighting current gaps in practice from a patient perspective. No members of the PPI representatives or wider PPI roundtable group were interviewed as participants in the present study described here.

## RESULTS

Drawing on 54 semistructured interviews across 9 different sites, in 4 countries (see online supplemental table 2). We identified five themes, influencing both workarounds and configuration: *safety and workarounds, evolution away from highly configurable and customisable solutions, vendor–client relationship, the role of governance* and *finding the 'sweet spot'*. Supporting quotes are provided for each theme, with additional quotes provided in online supplemental table 3.

Six of our fieldwork sites had opted to purchase an integrated commercial system, rather than maintaining their own home grown system, although two sites still employed the so-called 'Best of Breed' model (see online supplemental table 2).

## Safety and workarounds

As previous work has acknowledged, changes to the system are in many cases needed to allow functionality, which enables particular workflows. Changes were needed to reflect workflows beyond the North American context where the system had been developed. Barcode scanning of medicines, which is increasingly a safety feature included in many commercial systems, did not always match the labelling of available products with clear safety, which meant staff created a workaround.

> 'The biggest problem we have is when we can't scan products. In the beginning, we had a lot of troubles with that because as a nurse I scan the [label] but then I scan the antibiotics and the system says hey, I don't know these antibiotics. So I had to make a workaround …There were a lot of wards who had their own work around, and that's something we discovered in the medication commission committee, and we had to work at that, and then we discovered it was due to the barcode scanning'—Site C, vendor liaison nurse—Netherlands

Individuals improvised particular processes when they were not sure how to follow the 'standardised route'. (see quotation 2 Table 3: online supplemental material.)

In some cases, staff struggled to adjust when the system curtailed workarounds. While this increased safety, it meant staff could no longer resort to shortcuts to save time. (see quotation 3 Table 3: online supplemental material.)

## Evolution away from highly configurable and customisable solutions

A number of participants noted the drawbacks of a highly modifiable and flexible system, while accepting the benefits of taking a more cautious approach to modification. Several sites described how vendors had supported a high degree of local customisation or configuration in the early days of implementation. This was often the case when the vendor was trying to roll out their system in a new national or specialty context.

> '[COTS 1] very much supported us to, sort of, go out and build things out the way that'd work for us. And a lot of the decisions we made at the very start are things that are coming back and causing problems now…is that we went to every single clinic and asked them to design their own documentation. We made a decision that every single clinical trial would have its own set of orders. And so, now we have 4500 different power plans associated with clinical trials—chemotherapy plans—and it is impossible to maintain'—Site E, CMIO—US.

It later became clear to a number of sites as well as vendors that too much modification could lead to an unwieldy system creating extra work for the vendor. (see quotation 5 Table 3: online supplemental material.)

While the system now has the functionality to mimic the 'traditional drug chart view', this was considered less valuable than the speed and usability for all clinical areas, to which users had become accustomed. Those with responsibility for modifying the system need to balance the safety risks arising from a lack of system functionality with

potential over customisation. Experience of the specific needs of a particular site, coupled with familiarity with a vendor mean that staff are able to make more informed judgements about whether taking a configuration route was the best way to improve safety overall. (see quotation 6 Table 3: online supplemental material.)

### Vendor–client relationship

Several interviewees noted that making highly specific modifications of the system risk opening a gulf between the site and the vendor. While there is a view that sites are clients and have some rights to ask for what they want, a number of interviewees acknowledged that suppliers provide a service at a general level and that the sites can feed into that.

'I talk about influencing the shape of a product it doesn't necessarily mean that you can ask for a very specific bit of functionality that nobody else wants and you're going to get. But actually, as I say, you do have influence if you're doing interesting things that actually the broader health community are interested in. So you've got to constantly have a mind of the suppliers are out there to run their business and the thing that they're really interested in is maximising the reach of their product. So they really want to talk to you when you're doing things that are actually interesting in a broader context…if you drive customisation to a point where what you're doing is unique, you're setting yourself up to fail.'—Site A, CIO—UK.

Tailoring the system to a very high degree can also lead to individual sites having to take responsibility for ensuring that staff understand and can safely access the specific customised or configured functionalities. (see quotation 8 Table 3: online supplemental material.)

A mutual shaping of needs and vision was occurring in a number of sites, where key individuals had developed close working relationships with the vendors. Those staff selected to complete extensive training preimplementation would then move on to be 'super users'. This was a mechanism used as a strategy to bring existing workflows together with the system's capacities to avoid both workarounds and excessive customisation. (see quotation 9 Table 3: online supplemental material.)

### The role of governance

Many interviewees balanced a recognition that changes should be minimal with an acceptance that vendors cannot design systems to fit every context. This meant that governance and monitoring formed a significant part of how staff prioritised and retained particular configurations or customisations.

'So, it's important to have a structure, right. On the pharmacy side, we have a few different committees. We have an adult clinical committee, we have a paediatric committee, we have an oncology committee. So, any drug that we want to configure or optimise or modify really needs to be presented to this committee for ultimate approval. And we have a higher-level governance too.'—Site H, pharmacy manager—US.

Internal governance is necessary to consider both petitions for customisation before they are forwarded to the vendor and to prioritise configurations in relation to, for example, alert functions or protocols for specific processes. A data-led approach, enabled by system functionality, can be used to inform meetings with relevant professionals regarding prioritisation of particular configurations of the system.(see quotation 11 Table 3: online supplemental material.)

An interviewee with oversight of a number different hospital sites run by a single health provider, monitored and reversed configurations to, for example, alert systems, in order to enforce and support uniform expectations about safety functions. (see quotation 12 Table 3: online supplemental material.)

Simultaneously, data on any alert rule changes in local hospital sites were reviewed as a basis for prioritising those changes, which were useful at the local level. Staff also need to work towards an awareness that the system at any given time will have limitations and may not be doing everything that people think it does. (see quotation 13 Table 3: online supplemental material.)

### Finding the 'sweet spot'

Vendors can encourage greater conformity by ensuring that only technologies and practices that follow the rules can take advantage of the interoperability and integration opportunities offered by their products. One US-based CMIO made a comparison to the technology company Apple, which limits how its products are modified or used with other products.

'Apple will say, no, you have to follow our rules, and if your app doesn't follow these rules, we won't let you use it on our platform. The overall user experience is tighter and more consistent. I think, [COTS 2] is a bit like that…'—Site H, CMIO—US

In this case, the interviewee did not see the imposition of standards as negative but rather as contributing to a more 'consistent' experience for users. (see quotation 15 Table 3: online supplemental material.)

One strategy, explicitly encouraged by vendors in some cases, was for individual sites to have their needs met as part of a network of users. Similarly, in the following instance, where the site formed part of a larger healthcare organisation with one shared system, the was an emphasis placed on only pushing for those changes that could be enacted in every site. (see quotation 16 Table 3: online supplemental material.)

The ability to manage the tension between local and wider applicability of a system modification was referred to as finding 'a sweet spot' between 'out of the box functionality and configuring it.'

## DISCUSSION

This qualitative study involving digitally advanced hospitals highlights a number of important principles around customisation and configuration to be considered when optimising an ePrescribing system. Misalignment of functionality and organisational needs is seen as one reason why hospitals go down the customisation route.[4] Our data suggest that Epic and Cerner, which were designed in a North American context, required changes to functionality to capture workflows in other national settings. The differences in roles, workflows and policy imperatives for access to particular data sources in the different countries required an early rethink soon after implementation. Where sites dealt directly with the vendor, the hospital worked with the vendor to explain and introduce particular national requirements.

Workarounds continue to be a complex issue that require substantial consultation with providers. Workarounds may allow staff and systems to gradually coalesce[7] while avoiding the pitfalls of over customisation.[19] We found a cautious approach to configuration and customisation was accompanied by low tolerance for the long-term use of workarounds. There was a high degree of awareness about the potential safety and administrative drawbacks of by passing the system. Staff had also developed vigilance with regard to the limitations of the system and the need for caution about what safety functions are actually enabled at any given time.

The mismatch between workflow and system functionality has been combatted in a number of sites by the "super users" who are deployed to encourage staff to work within the systems capacities. Sites had learnt over time to be more adaptable in accepting the way that they balance between their own and vendors' needs. In some cases, key staff members developed a close relationship with the vendor, allowing them to maintain a current appreciation of the potentialities of the system.

Innovative forms of monitoring and testing the effectiveness of particular configurations were evident, with one site in particular employing data to monitor acceptance and use of particular alerts. This data-driven approach was combined with staff oversight and comparison with the uptake of similar alerts across other sites. This allowed rigorous monitoring and prioritisation work to balance beneficial and helpful optimisation against the drawbacks of making multiple changes to the system. Drawbacks included increasing responsibility on the sites for training, maintenance and improvement.

Finding the 'sweet spot' appeared to involve a subtle evolution away from a demand for highly configurable and customisable solutions and towards finding solutions that could be up scaled. Sites were apt to search for allies within the national system user network to lobby for or finance requested changes to the system. Therefore, in order to fix a mismatch between functionality and workflow, sites would need to be proactive in finding other users experiencing similar problems. There appeared to be an increasing willingness for sites to encourage behaviour change in staff and to scale back their requests for changes to the system driven by both safety and usability concerns.

### Strengths and limitations

This is one of the first studies to investigate the impact of ePrescribing optimisation attained by customisation and configuration practices. The case studies presented provide insight into diverse regional and national contexts. The focus on optimisation enables the sharing of key learning in the interactive evolution of users and systems. By focussing on professionals within advanced sites, we have been able to capture the benefits of their experiences post implementation.

Due in part to the disruption caused during fieldwork by COVID-19, we did not collect equal levels of data in all sites. This was mitigated wherever possible by carrying out longer interviews. Hospitals using two integrated systems or COTS (Epic and Cerner) are dominant within the sample, best-of-breed systems are far less represented. The US and UK contexts are over represented compared with the Netherlands and Norway. The focus on OECD member countries narrowed the range of hospitals we could include.

### Interpretation in the light of the wider published literature

Customisation has been presented as giving positive outcomes for users and patients.[21] It is also inevitable given the difficulty of capturing workflows without knowledge of the environment and actual challenges faced by staff.[8] While the system may be designed to reshape practices in a more efficient and safer way, initially there is often misalignment between what is required by the specialisation or organisation and the functionality of the system.[4 10] Vendors worked closely with early adopters bringing a lot of specialised knowledge about national settings and specialities to the relationship, which our data suggest enabled customisation. This appeared to become less desirable not only to vendors but more recently also to users, over time. Hospital sites are increasingly willing to adopt a parsimonious approach to configuration and customisation.

The mutual learning between site and vendor is an ongoing process requiring staff and resourcing. Where systems are developed in a distinct national context, a network of other users may be sought out in developing necessary functionality at a quicker pace. More recent studies have pointed to greater awareness of the balance of costs and benefits in relation to extensive modifications, including wide variation between sites using the same system.[4 22] Future research could look at the ways in which relationships between vendor and individual sites are shaped by the presence of a critical mass of users in a given national context.

## CONCLUSION

Our data suggests an increasing acceptance from interviewees that the needs of individual sites would be met as

part of a network of users potentially of the same product. While some frustrations with delays on the part of the vendor to changes required by sites remained, there was little enthusiasm for making too many changes at a local level. Sites acknowledged the danger of becoming too responsible for an extremely bespoke system. Simultaneously, interviewees were cognisant that the vendor would not foresee all eventualities, especially in specialities or within scientifically as well as digitally advanced hospitals. However, they had learnt the benefit of considering broad applicability of optimisations.

**Acknowledgements** We thank all the participating sites and the individuals who participated in an interview for this study. We thank Serena Tricarico, Kieran Turner, Toni Wigglesworth, and our Patient and Public Involvement representatives, Antony Chuter and Jillian Beggs, for their support and feedback throughout the project. We also acknowledge the support of colleagues from the Department of Health and Social Care, the National Health Service and the Medicines and Healthcare products Regulatory Agency: Ann Slee (NHS), Jason Cox (DHSC), Richard Cattell (NHS), Helen Causley (DHSC), Paul Stonebrook (DHSC), Mick Foy (MHRA), Kathryn Ord (MHRA), and Graeme Kirkpatrick (NHS). We thank the referees for reviewing this manuscript.

**Contributors** AS conceptualised the project and designed the study. CH recruited study sites. CH and SM recruited individual participants within study sites and conducted data collection. CH prepared the first draft of the manuscript. All authors contributed to the final draft of the manuscript and approved it for submission. AS is the guarantor for this study.

**Funding** This study/project is funded by the National Institute for Health Research (NIHR) (Optimising ePrescribing in Hospitals (PR-ST-01-10001)/Policy Research Programme).

**Disclaimer** The views expressed are those of the author(s) and not necessarily those of the NIHR or the Department of Health and Social Care.

**Competing interests** None declared.

**Patient and public involvement** Patients and/or the public were involved in the design, or conduct, or reporting, or dissemination plans of this research. Refer to the Methods section for further details.

**Patient consent for publication** Not applicable.

**Ethics approval** This study was granted ethical approval by the Usher Research Ethics Group (University of Edinburgh) on 21/01/2020 (ref.1906), and the relevant NHS research and development approvals were acquired for UK-based sites on 23/01/2020 (ref.19/HRA/7015). As this research is exclusively being conducted with healthcare professionals, there is no involvement of vulnerable groups. As we are focusing this research on well-known, world-leading institutions in health informatics, extra care should be taken when presenting research findings to not compromise the anonymity of the respondents. Specific job titles have been masked when used in conjunction with individual quotes, and care has been taken to avoid any quotes that could be used to potentially identify individuals. All audio files, transcripts and consent forms will be kept in a secure, password-protected folder within the University of Edinburgh's DataStore (for further information please consult the protocol for this study (17)).

**Provenance and peer review** Not commissioned; externally peer reviewed.

**Data availability statement** Data are available from the corresponding author upon reasonable request.

**ORCID iDs**
Catherine Heeney http://orcid.org/0000-0002-0725-974X
Stephen Malden http://orcid.org/0000-0002-5819-6347

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
