## [Reviewer comments · BMJ Open]

ARTICLE DETAILS

TITLE (PROVISIONAL)	Finding the 'sweet spot' between customisation and workflows when optimising ePrescribing systems: A multisite qualitative study
AUTHORS	Heeney, Catherine; Malden, Stephen; Sheikh, Aziz

VERSION 1 – REVIEW

REVIEWER	Karen McBride-Henry Victoria University of Wellington, School of Nursing, Midwifery & Health Practice
REVIEW RETURNED	07-Jul-2022

GENERAL COMMENTS	Thank you for the opportunity to review this interesting manuscript. I appreciate the significant effort the research team have made in gathering the interviews and analysing the transcripts. I offer the following suggestions that I hope will help strengthen the manuscript: 1. Keywords do not describe the research that has been conducted. The MeSH terms could be used to expand the keywords.2. Abstract – The authors need to include additional information about the analysis method in the abstract.3. Introduction: p 5, line 6 – The responsibility for ongoing maintenance, once customisation occurs, should also be mentioned in the context of this paragraph.4. P 5, line 16. The first sentence of the paragraph should be re-written.5. P 5, line 32. The authors state optimisation strategies are "still relatively unexplored"; this phrase also appears in the abstract. What do the authors mean by the term 'relatively'? Please include additional information, cite the relevant literature sources on this topic, and review the abstract accordingly.6. Methodology:a. The research aim in the manuscript differs from the one presented in the abstract; this needs to be corrected.b. How were the questions created for the semi-structured interview? Please provide examples. The supplementary material on the interview guide was not available.c. How were the specific participants identified? Information is provided on the hospital selection process but not for individual participants.d. How was consent for participation gained? What role did the participants have in supporting the analysis? There was a Patient and public involvement statement, but additional details on the findings' trustworthiness are warranted.7. Placing all quotes in Table 3 made reading the manuscript disjointed. In reality, readers would not necessarily refer to the narratives, which undermines the quality and intent of qualitative research. I appreciate that word counts are an issue for most
---

	journals, but I strongly encourage the authors to nest the narratives in the manuscript. 8. The findings, in keeping with the analysis approach, are descriptive but provide insights into how people manage the optimisation of an ePrescribing system. The discussion section reflects the study findings, and recommendations for practice and future research are explored. 9. P 16, line 4 should read COVID-19. Thank you again for the opportunity to review this manuscript. I wish the research team the best with their future research projects.
--	---

REVIEWER	Jude Heed Newcastle University
REVIEW RETURNED	19-Jul-2022

GENERAL COMMENTS	Overall, the paper is very interesting and topical and addresses a gap in the current literature. It is well written but there are some typographical errors. In the introduction (paragraph 3 line 3) two words have been combined "routcan" and later (paragraph 4 line 1) the sentence "modifications to reflect to mediate local requirements" does not read well, it may be preferable to use reflect or mediate but not both. The abstract is quite vague and would benefit from more detail regarding the methods employed and the emerging themes from data analysis. The introduction is relevant and informative, however when exploring configuration versus customisation (paragraph 2) I wondered if financial aspects may have been considered as a potential reason for local customisation. The results and discussion are very interesting and balanced. I noted that one particular quotation (quotation 3 site C) refers to the identification of illegal prescribing practices which cannot and should not be replicated in EP systems however this is not really unpacked in the discussion. Overall an interesting and well written paper, which was enjoyable to read.
--

VERSION 1 – AUTHOR RESPONSE

Thank you for the opportunity to review this interesting manuscript. I appreciate the significant effort the research team have made in gathering the interviews and analysing the transcripts.	Thank you for taking the time to review our manuscript and provide feedback.
1. Keywords do not describe the research that has been conducted. The MeSH terms could be used to expand the keywords.	Thank you- we have now added the following keywords that we feel better capture the content of this manuscript: “Quality in health care” and “Adverse events”. We have deleted “public health”.
2. Abstract – The authors need to include additional information about the analysis method in the abstract.	We have now added the following to the methods section of the abstract: Interview transcripts were analysed using an

	inductive thematic analysis.
3. Introduction: p 5, line 6 – The responsibility for ongoing maintenance, once customisation occurs, should also be mentioned in the context of this paragraph	We have now adapted the final sentence of the paragraph to reflect the issue of ongoing maintenance post customisation: However, drawbacks of extensive configuration and customisation include being left behind for standard updates from the vendor and uncertainty as to where responsibility for making improvements and routine maintenance resides after customisation of a system has taken place (9, 10).
4. P 5, line 16. The first sentence of the paragraph should be re-written.	We have now restructured this sentence to read more fluently: Modifications to reflect local requirements and system capabilities are key to system optimisation (4, 7, 15), which has been defined as...
5. P 5, line 32. The authors state optimisation strategies are "still relatively unexplored"; this phrase also appears in the abstract. What do the authors mean by the term 'relatively'? Please include additional information, cite the relevant literature sources on this topic, and review the abstract accordingly.	At present, the majority of the literature on ePrescribing has focused on the implementation of new systems to replace paper-based prescribing, rather than the optimisation of existing systems (although, the two terms are somewhat difficult to define in isolation from each other). Our use of the term "relatively" therefore tries to highlight this, however we agree with the reviewer that some context would be helpful to the reader. We have now added to this phrase to highlight that we mean relative to the implementation literature: The impact and learning arising from optimisation strategies, reliant on configuration and customisation, over time is still relatively unexplored in existing literature on ePrescribing in comparison to the equivalent literature on the implementation of new ePrescribing systems to replace paper-based practices.
6. Methodology: a. The research aim in the manuscript differs from the one presented in the abstract; this needs	a. We have now amended the aims in the methods so that it matches what is stated in the abstract.

to be corrected. b. How were the questions created for the semi-structured interview? Please provide examples. The supplementary material on the interview guide was not available. c. How were the specific participants identified? Information is provided on the hospital selection process but not for individual participants. d. How was consent for participation gained? What role did the participants have in supporting the analysis? There was a Patient and public involvement statement, but additional details on the findings' trustworthiness are warranted.	b. Apologies that the supplementary material was not available. We now include some details of the questions asked during interviews in the data collection section that are specific to customisation/configuration that this paper relates to. (Specifically, participants were asked to detail any customisations/configuration that had taken place at their sites, and to summarise the reasons for this, and the perceived benefits/repercussions of such optimisations on the overall functionality of the ePrescribing systems) Interview questions were developed through insights gathered during initial “expert roundtable” discussions with ePrescribing experts prior to data collection. The research team then refined the questions and piloted them internally. We now briefly describe this step in the “data collection” section of the manuscript. c. We have now added the following text to explain the identification of potential participants: In all cases, we had an initial meeting with a gatekeeper(s) who helped identify relevant professionals who had been expensively involved in their respective ePrescribing system optimisations. d. We now provide additional details of the consent process in the data collection section: If participants were willing to participate, they returned a signed and dated consent form via email to the corresponding researcher(s), who then arranged with them an appropriate time to conduct the interview. We also provide further context on the role patient and public involvement played in supporting the analysis. Specifically, participants in the interviews had no further involvement as PPI representatives, as this was a separate group of stakeholders. We have added the following text to the PPI section to highlight this:
--	--

	No members of the PPI representatives or wider PPI roundtable group were interviewed as participants in the present study described here.
7. Placing all quotes in Table 3 made reading the manuscript disjointed. In reality, readers would not necessarily refer to the narratives, which undermines the quality and intent of qualitative research. I appreciate that word counts are an issue for most journals, but I strongly encourage the authors to nest the narratives in the manuscript.	We agree with the reviewer that qualitative findings are far better interpreted when nestled within the main body of text. The supplementary table was used for word count purposes as pointed out. However, considering the importance of the quotes in illustrating the research findings, we have now nestled one quote for each theme/sub-theme within the text, referring the reader to the table for additional insights.
8. The findings, in keeping with the analysis approach, are descriptive but provide insights into how people manage the optimisation of an ePrescribing system. The discussion section reflects the study findings, and recommendations for practice and future research are explored.	Thank you
9. P 16, line 4 should read COVID-19.	We have now amended this accordingly.
Overall, the paper is very interesting and topical and addresses a gap in the current literature. It is well written but there are some typographical errors. In the introduction (paragraph 3 line 3) two words have been combined "routcan" and later (paragraph 4 line 1) the sentence "modifications to reflect to mediate local requirements" does not read well, it may be preferable to use reflect or mediate but not both.	Thank you for taking the time to review our manuscript and provide feedback. We have now rectified the typos and grammatical errors highlighted by the reviewer in the manuscript.
The abstract is quite vague and would benefit from more detail regarding the methods employed and the emerging themes from data analysis	We have added additional information in the abstract regarding the methods and the themes identified.
The introduction is relevant and informative, however when exploring configuration versus customisation (paragraph 2) I wondered if financial aspects may have been considered as a potential reason for local customisation.	Thank you for this suggestion, although this did not explicitly come up in the data we collected, there were overarching financial reasons why customisation was sometimes preferred (out of necessity) to the implementation of a newer/more expensive system in some of our study sites. We have now highlighted the potential influence of financial influences on customisation practices in

	the introduction with the following: Hospitals resort to this where the vendor does not provide sufficient configurability to capture necessary workflows, or where it is more financially viable to configure/customise existing systems than it is to purchase additional systems (10).
The results and discussion are very interesting and balanced. I noted that one particular quotation (quotation 3 site C) refers to the identification of illegal prescribing practices which cannot and should not be replicated in EP systems however this is not really unpacked in the discussion.	Thank you for the positive comments. Regarding this particular quote, We appreciate that this highlights a specific breach of prescribing laws (in this case for the Netherlands, but is applicable to the UK and US too), that highlights the wider issues this paper seeks to address. However, Given the wider focus of the paper is on the role customisation plays in reducing the use of improper workarounds in general (whether they be unlawful or just impractical), we feel that focusing on individual examples in the discussion would be difficult, and have taken a more general, overview approach with summarising the findings. We do however feel that such issues deserve more attention and research- specifically around how to reduce such practices.
Overall an interesting and well written paper, which was enjoyable to read.	Thank you again to both reviewers for taking the time to review and comment on our manuscript.

VERSION 2 – REVIEW

REVIEWER	Karen McBride-Henry Victoria University of Wellington, School of Nursing, Midwifery & Health Practice
REVIEW RETURNED	05-Oct-2022

GENERAL COMMENTS	Thank you for the opportunity to review your revised manuscript. I appreciate your time and energy in addressing my previous comments. One minor editorial issue that I noted that should be adjusted is that supplementary Table 3 should be re-named as it appears before supplementary Table 2 in the text of the manuscript. I appreciated the inclusion of narratives within the body of the manuscript, which facilitates an enhanced understanding of the nuances related to the findings. Well done on completing a unique and novel piece of research.
---